# Nuclear Matter in 1 + 1 Dimensions

**Robert D. Pisarski** [1,*] , **Marton Lajer** [2], **Alexei M. Tsvelik** [2] and **Robert M. Konik** [2]

[1] Department of Physics, Brookhaven National Laboratory, Upton, NY 11973, USA

[2] Division of Condensed Matter Physics and Material Science, Brookhaven National Laboratory, Upton, NY 11973, USA; lajer.marton@wigner.hu (M.L.); tsvelik@bnl.gov (A.M.T.); rmk@bnl.gov (R.M.K.)

[*] Correspondence: pisarski@bnl.gov

**Abstract:** We review the solution of QCD in two spacetime dimensions. Following the analysis of Baluni, for a single flavor, the model can be analyzed using Abelian bosonization. The theory can be analyzed in strong coupling, when the quarks are much lighter than the gauge coupling. In this limit, the theory is given by a Luttinger liquid.

**Keywords:** dense QCD; quarkyonic; Luttinger liquid





## 1. Introduction

It is a pleasure for us to be able to contribute to this volume in memory of Dr. Hector de Vega. Hector was both a great physicist and a true gentleman. The subject of this pedagogical article is not directly about keV warm dark matter, to which this volume is devoted, but to other subjects on which Hector worked before delving into cosmology, in particular his work on integrable models in 1 + 1 dimensions.

In particular, R.D.P. would like to express that it is a great honor to contribute to this volume, as I had both the joy of working with Hector [1,2], and of considering him a good friend.

The subject of this paper is the behavior of quantum chromodynamics (QCD) in 1 + 1 dimensions. This problem has been analyzed several times over the years, and recently [3]. The purpose of this article is to bring together the results, which tend to span a rather wide range of methods. What is interesting is that if one concentrates just on the low energy excitations for cold, dense QCD in 1 + 1 dimensions, then the theory reduces to that for a single, massless boson, which propagates with a speed less than that of light. This is what is known as a Luttinger liquid.

We first sketch how to derive these results, and then conclude with some speculations as to their possible relevance for cold, dense QCD in 3 + 1 dimensions.

## 2. QCD for a Single Flavor

We begin with the usual Lagrangian for QCD, where the quarks lie in the fundamental representation of a $SU(N_c)$ color group,

$$L = \int d^2x \left[ -\frac{1}{4} \mathrm{tr} G^{\mu\nu} G_{\mu\nu} + \bar{q}_{f,\sigma} \gamma^\mu D_{\mu,\sigma\sigma'} q_{f,\sigma'} + m \bar{q}_{f,\sigma} q_{f,\sigma} \right] . \tag{1}$$

$$D_\mu = \partial_\mu - ig A_\mu \;\; ; \;\; G_{\mu\nu} = \partial_\mu A_\nu - \partial_\nu A_\mu - ig[A_\mu, A_\nu] . \tag{2}$$

The gauge coupling $g$ has dimensions of mass in two spacetime dimensions; the quark fields $\bar{q}, q$ carry $a, b \ldots = 1, \ldots N_f$ flavor and $\alpha, \beta \ldots = 1, \ldots N_c$ color indices.

In two dimensions, gauge fields are not propagating degrees of freedom, which allows one to simplify the theory. It helps to choose the gauge $A_0 = 0$. This still leaves $A_x$, which, for simplicity, we denote just as the color matrix $A$. Baluni [4] noted that one can further

choose a "hybrid" gauge, which vastly simplifies the analysis. First, to assume that the gauge potential, $A$, is an off-diagonal matrix, and that the electric field, $E$, is diagonal:

$$A^{\alpha\beta} = 0 \; , \; \alpha = \beta \; ; \; E^{\alpha\beta} = 0 \; , \; \alpha \neq \beta \; ; \; E^{\alpha\alpha} = -\frac{1}{2}\left(e^{\alpha} - \frac{1}{N_c}\sum_{\beta=1}^{N_c} e^{\beta}\right) . \qquad (3)$$

This gauge is useful in imposing Gauss' law, $D_x E = J_0$, where $J_0$ is the quark current. Besides the contribution of the quark current, this also involves the covariant derivative, $D_x$, and so, the commutator of the gauge potential with the electric field. In the hybrid gauge, however, the diagonal elements of the electric field are directly proportional to the diagonal elements of the quark current, while the off-diagonal elements of the gauge potential are proportional to the off-diagonal elements of the quark current:

$$\partial_x e^{\alpha} = j_0^{\alpha\alpha} \; ; \; ig(e^{\alpha} - e^{\beta})A^{\alpha\beta} = j_0^{\alpha\beta} \; \alpha \neq \beta \; ; \; j_0^{\alpha\beta} = \bar{q}^{\alpha}\gamma_0 q^{\beta} . \qquad (4)$$

There is no sum over repeated indices: $j_0^{\alpha\alpha}$ is just a single element of the diagonal quark current, with color $\alpha$. For the color diagonal current, $A^{\alpha\beta}$ does not enter, because it is taken to be purely off-diagonal. Similarly, for the elements of the color current which are off-diagonal in color, the spatial derivative of the electric field does not enter, because it is assumed to be purely diagonal.

In two dimensions, there is no magnetic field, so the action for the gauge field just involves the square of the electric field. Thus the above doesn't look very useful, since $j_0$ is proportional to the spatial derivative of the electric field. This is where bosonization is useful, as the current $j_0 \sim \partial_x \phi$, where $\phi$ is a boson field. By Gauss' law, in the hybrid gauge, the electric field $e^{\alpha}$ is naturally proportional to the scalar field of bosonization.

The result for the Hamiltonian after bosonization is

$$\begin{aligned}
\mathcal{H} &= \mathcal{H}_0 + \mathcal{H}_{\text{int}} \; , \\
\mathcal{H}_0 &= \frac{1}{2}\sum_{\alpha=1}^{N_c} \pi_{\alpha}^2 + 2m\Lambda\left(1 - \cos(2\sqrt{\pi}e^{\alpha})\right) , \\
\mathcal{H}_{\text{int}} &= \frac{g^2}{8\pi N_c}\sum_{\alpha,\beta=1}^{N_c}\left(e^{\alpha} - e^{\beta}\right)^2 + \Lambda^2\sum_{\alpha,\beta=1}^{N_c}\frac{\sin(2\sqrt{\pi}(e^{\alpha} - e^{\beta}))}{(e^{\alpha} - e^{\beta})} ,
\end{aligned} \qquad (5)$$

where $\pi^{\alpha}$ is the momentum conjugate to the electric field $e^{\alpha}$. We are sloppy about normalization, and in particular, about normal ordering. As is typical with bosonization, most terms contain ultraviolet divergences (from tadpole-like diagrams), and are only well defined if normal-ordered. The terms in $\mathcal{H}_0$ are standard for bosonization. The term in $\mathcal{H}_{\text{int}}$, which is proportional to $(e^{\alpha} - e^{\beta})^2$, is related to the usual electric field term. The second term in $\mathcal{H}_{\text{int}}$ arises from the current–current interaction which the gauge field induces. The mass scale $\Lambda$ arises from normal ordering, and previous analysis took it as proportional to the gauge coupling, $\Lambda \sim g$. While this at least ensures that the perturbative expansion is well defined, it is not evident that this is consistent. In particular, it is not obvious how to normal order expressions such as $\sin(\phi)/\phi$.

This form of the theory has been analyzed by several authors [5–12]. We assume a single flavor, because with two or more flavors, $\mathcal{H}_{\text{int}}$ involves the conjugate momenta, $\pi^{\alpha}$, as well as the coordinates $e^{\alpha}$ [5–8]. This significantly complicates the analysis. For several flavors, it is more useful to adopt non-Abelian bosonization [3,7,8,10–13] (for a recent review of non-Abelian bosonization, see Ref. [14]). Even so, the case of a single flavor is still most illustrative.

Returning to the present approach, even with a single flavor, there are $N_c - 1$ coupled sine-Gordon models, with a peculiar coupling, from the last term of $\mathcal{H}_{\text{int}}$. A single sine-Gordon model has a rich spectrum of excitations: both small fluctuations, analogous to mesons, and kinks and anti-kinks, analogous to baryons and anti-baryons. With $N_c - 1$ coupled sine-Gordon models, the spectrum becomes even more convoluted. Notice, how-

ever, that the fields for the lightest mesons are naturally proportional to the $\sim e^\alpha - e^\beta$. From the above, their mass is $\sim g$, and so this represents a set of mesons/glueballs. There are then baryons, given by kinks [5–8]. For further studies of the spectra of this model, see Refs. [3,8–12]. Certainly, as a confining gauge theory, it is expected that all excitations are massive.

In Ref. [3], an alternate approach was taken. The gauge $A_x = 0$ was taken, and the free gauge field integrated out:

$$
\begin{aligned}
H &= \sum_{\alpha=1}^{N_c} \int dx \left[ -i\bar{q}_{R,\alpha}\partial_x q_{R,\alpha} + i\bar{q}_{L,\alpha}\partial_x q_{L,\alpha} - m(\bar{q}_{R,\alpha}q_{L,\alpha} + \bar{q}_{L,\alpha}q_{R,\alpha}) \right] \\
&\quad - \pi g^2 \int dx dy \sum_{A=1}^{N_c^2-1} J_0^A(x)|x-y|J_0^A(y) ;
\end{aligned}
\tag{6}
$$

where $q_{L,R} = (1 \pm \gamma_5)q$ are the right- and left- moving components of the quark field. The currents for right- and left- moving quarks, $J_{R,L}^A$, are given by

$$
J_0^A = J_R^A + J_L^A \;,\quad J_R^A = \bar{q}_{R,\alpha}\,(t)_{\alpha\beta}^A\,q_{R,\beta},\; J_L^A = \bar{q}_{L,\alpha}\,(t)_{\alpha\beta}^A\,q_{L,\beta} \,,
\tag{7}
$$

With $(t^A)_{\alpha\beta}$ matrices for the adjoint representation. The chiral currents $J_{R,L}$ obey a $SU(N_c)$ Kac–Moody algebra [14].

Introducing the chiral currents is especially useful when expanding about the massless limit. With the fields $q_{R,\alpha} \sim \exp(i\sqrt{4\pi}\varphi_\alpha)$ and $q_{L,\alpha} \sim \exp(i\sqrt{4\pi}\bar{\varphi}_\alpha)$. After bosonizing the current-current interaction, one obtains two terms in the potential. The first is from the diagonal elements,

$$
V_{\text{Cartan}}(x) = -\pi g^2(1-1/N_c) \int dy |x-y|\partial_x \Phi_\alpha(x)\partial_y \Phi_\alpha(y) = g^2(1-1/N_c)\Phi_\alpha(x)\Phi_\alpha(x) ,
\tag{8}
$$

where $\Phi = \varphi + \bar{\varphi}$. The off-diagonal terms contribute

$$
\begin{aligned}
V_{\text{off-diag}} &= \sum_{\alpha>\beta} \int dy \frac{g^2}{4\pi}|x-y|^{-1}\Big\{ \cos[\sqrt{4\pi}[\varphi_{\alpha\beta}(x) - \varphi_{\alpha\beta}(y)]] \\
&\quad + \cos[\sqrt{4\pi}[\bar{\varphi}_{\alpha\beta}(x) - \bar{\varphi}_{\alpha\beta}(y)]] - 2|x-y|^2\cos[\sqrt{4\pi}[\varphi_{\alpha\beta}(x) + \bar{\varphi}_{\alpha\beta}(y)]]\Big\}
\end{aligned}
\tag{9}
$$

With $\varphi_{\alpha\beta} = \varphi_\alpha - \varphi_\beta$.

It is useful to contrast these analyses with the solution of QCD at large $N_c$ by 't Hooft [15]. Again, one goes to axial gauge, so that the propagator for the gauge field reduces to that of a free field. Doing so, it is possible to solve the Schwinger–Dyson equation for the quark propagator. This demonstrates a confining spectrum. This has been extended to nonzero quark density by Bringoltz [16], who finds chiral density waves for a massive quark.

There is a peculiarity in the diagonal potential, $V_{\text{Cartan}}$. The bosonized fields, $\varphi$ or $\bar{\varphi}$, are manifestly periodic. However, $V_{\text{Cartan}}$ is clearly not periodic. This is also present in the previous form, the "mass" term in Equation (5). It was not apparent in this form, nor the lack of periodicity appreciated. In terms of the $\varphi$ and $\bar{\varphi}$ fields, though, it demonstrates that since periodicity is lost, the only topologically non-trivial configurations are those which involve the sum of all angles. The only soft mode remaining is the sum of the $\Phi_\alpha$'s,

$$
\Phi = \frac{1}{\sqrt{N_c}} \sum_\alpha \Phi_\alpha .
\tag{10}
$$

This is completely unaffected by the potential terms in Equations (8) and (9), which only involve the differences, $\varphi_\alpha - \varphi_\beta$, etc.

The ground state corresponds to the state where all fields are equal. Projecting the mass term onto this vacuum gives a single sine-Gordon model:

$$\mathcal{H}_{eff} = \frac{1}{2}\left\{\Pi^2 + (\partial_x\Phi)^2\right\} + 2\frac{\tilde{m}}{2\pi}\left[1 - \cos\left(\sqrt{\frac{4\pi}{N_c}}\Phi\right)\right]. \tag{11}$$

In this expression, $\tilde{m}$ is proportional to the original mass scale, including the effects of renormalization by normal ordering. Naturally, the projection assumes that the energy scale generated by the mass term is much smaller than the energies of the mesonic fields. Besides the $U(1)$ field $\Phi$, there are also color singlet excitations above $\Lambda_{QCD}$, involving fluctuations of individual fields $\Phi_\alpha$ around the minimum of the potential. By going to energies below the scale of the gauge coupling, all of these massive degrees of freedom can be ignored.

This implies that in 1 + 1 dimensions, dense QCD is much simpler than one might expect. By bosonization, a nonzero chemical potential is incorporated simply by shifting

$$\left(\frac{4\pi}{N_c}\right)^{1/2}\Phi \to 2\,k_0 x + \left(\frac{4\pi}{N_c}\right)^{1/2}\Phi \;,\;\; k_0 = \frac{\mu}{N_c}\;. \tag{12}$$

This follows directly because $j_0 \sim \partial_x\phi$. It is only $\Phi$ that is affected, as fermion number only couples to the global $U(1)$ symmetry for fermion number. The resulting effective Lagrangian is then

$$\mathcal{L} = \frac{1}{2}(\partial_\mu\Phi)^2 - \frac{\tilde{m}}{2\pi}\cos\left(\sqrt{\frac{4\pi}{N_c}}\Phi + 2k_0 x\right), \tag{13}$$

As mentioned above, in vacuum, the spectrum of this model consists of a soliton with mass $m_s$, and anti-soliton with the same mass, and $2N_c - 2$ breathers, which are also massive. These are all gauge invariant states.

The chemical potential does not affect the system until $\mu > m_s$. At that point, the solution becomes massless, while all other states, the anti-solition and the breathers, remain massive. Analysis shows that at $\mu > m_s$, this model renormalizes into that of a Luttinger liquid:

$$\mathcal{L}_{eff} = \frac{\tilde{K}(\mu)}{2}\left[v_F(\mu)^{-1}(\partial_\tau\Phi)^2 + v_F(\mu)(\partial_x\Phi)^2\right], \tag{14}$$

This is a single, massless boson, which propagates with a speed less than that of light.

The solution of the model is the following. The overall normalization of the effective Lagrangian is the Luttinger parameter, $\tilde{K}$, while $v_F$ is the Fermi velocity; both are functions of the chemical potential, $\mu$. The extraction of these dependencies requires the exact solution of the sine-Gordon model.

The limits of these parameters are easy to understand. At the edge of the Fermi surface, where $k_F \to 0$, the Luttinger parameter $\tilde{K} \to 1$. In contrast, the Fermi velocity $v_F \to 0$. This implies that the $\Phi$ field does not propagate, as the spatial term vanishes. For asymptotically high density, $k_F \to \infty$, the Luttinger parameter $\tilde{K} \to 1/N_c$, and the Fermi velocity $v_F \to 1$. This implies that in the limit of infinite $N_c$, that there is a density at which $\tilde{K}$ goes from being of order one, as is typical of dilute fermions, to order $\sim 1/N_c$. This is only valid at infinite $N_c$; at finite $N_c$, $\tilde{K}$ varies smoothly with density.

The solution for arbitrary chemical potential can be carried out through the thermodynamic Bethe ansatz. This is valid for arbitrary fermion mass. It is also possible to compute using perturbation theory in the mass parameter. Details, and a solution for two flavors and colors, are given in Ref. [3].

For an arbitrary number of flavors, presumably the theory is always a Luttinger liquid. The solution is rather more complicated, and involves non-Abelian bosonization. For an arbitrary number of flavors, it is not direct to solve the theory in weak coupling, when the mass is much larger than the gauge coupling. Thus, it is only a conjecture that the theory is a Luttinger liquid, although in two dimensions, it is most natural to expect.

It is an extraordinary feature of Fermi surfaces in two dimensions that the excitations near the Fermi surface are not fermions, but bosons. This is only possible because of bosonization in two dimensions.

It is impossible to resist speculating upon whether something analogus happens in $3 + 1$ dimensions. In a quarkyonic phase [17–33], while the free energy is that of deconfined quarks and gluons, excitations near the Fermi surface are confined. It is conceivable that this introduces a strong anisotropy into the system, so that it is essentially one-dimensional. If true, then there is a complicated pattern of excitations which arise. Especially with two or more light flavors, an involved pattern of chiral density waves can arise.

What is most intriguing, however, is whether the quarkyonic phase in $3 + 1$ dimensions is a Luttinger liquid. That is, a type of non-Fermi liquid. In particular, are the excitations near the Fermi surface not controlled by nucleons, but by (effective) bosons. The properties of an effective non-Fermi liquid can be described by an (anisotropic) effective Lagrangian, and used to compute the transport properties of a quarkyonic regime.

**Author Contributions:** Investigation, A.M.T., M.L. and R.M.K.; Writing—review & editing, R.D.P. All authors have read and agreed to the published version of the manuscript.

**Funding:** The work of R.D.P. was supported by the U.S. Department of Energy under contract number DE-SC0012704; by B.N.L. under the Lab Directed Research and Development program 18-036; and by the U.S. Department of Energy, Office of Science, National Quantum Information Science Research Centers, Co-design Center for Quantum Advantage (C2QA) under contract number DE-SC0012704. The of M.J., A.M.T., and R.M.K. was supported by the U.S. Department of Energy, Office of Science, Materials Sciences and Engineering Division under contract DE-SC0012704.

**Conflicts of Interest:** The authors declare no conflict of interest.

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
