# Peer review of "Nuclear Matter in 1 + 1 Dimensions"

_universe, doi:10.3390/universe7110411_

Round 1

Reviewer 1 Report

The paper should be published after the misprints are corrected. Serious misprints are in line 63 and in eq.(6).

Author Response

I thank the referees for their comments, the type in line 63 was corrected.  In eq 6, I also used the same notation for the quark fields as in eq 1, and corrected the typo.

Reviewer 2 Report

The text gives a short overview of older significant results, and I suggest its acceptance pointing out, at the same time, a few misprints and inconsistencies:

  1. The text has four authors while starts with "I review".
  2. line 24: "... propagates with a speed of light less then one" (the speed is always taken to be one in dimensionless units) → "propagates slower than the speed of light which is taken to be one" or similar. Also in line 91.
  3. Lines 45, 63: misprints/torn sentences. 
  4. The very end of line 95: LaTeX error.
  5. Eq.(8): arguments under the integral, x, y, to make it consistent with the notations of arguments in Eq.(9)? The boundary terms, can they spoil the result?
  6. Line 81: a word is missing: "The is completely ...".
  7. Line 69 starts from a Capital Letter.
  8. In the middle of Line 82: Gordon is a name, Capital letter.
  9.  Eq.(11): a comment on the definition of {\tilde m} would be appreciated. 

Author Response

I thank the referee for their detailed and careful comments.  I have corrected all typos, including "I" to "we", and the confusing comments about speed.

As for point 5, it isn't a boundary term, I write things as functions of "x" to make that clear.

Point 9: I added a sentence about \tilde{m} after the equation.  In general the relation is involved, which I mention. 

Reviewer 3 Report

The paper is interesting and sound and I recommend publication. However, there are some minor points that should be addressed:

-Since Universe is not a condensed matter journal, I would suggest reviewing or including some references introducing the concept of bosonization. 

There are a few typhos:

-"ey" in line 45.

-"th" in line 63

-A point is missing before "The only", in the line before eq. (10).

-In the paragraph between lines 95 and 96, there is a typho at the end.

Author Response

I thank the referee for their comments, all typo's were changed, and a recent review to non-Abelian bosonization included, Ref. 15.